# Herb–Drug Interactions: Worlds Intersect with the Patient at the Center

**DOI:** 10.3390/medicines8080044

**Published:** 2021-08-05

**Authors:** Mary Beth Babos, Michelle Heinan, Linda Redmond, Fareeha Moiz, Joao Victor Souza-Peres, Valerie Samuels, Tarun Masimukku, David Hamilton, Myra Khalid, Paul Herscu

**Affiliations:** 1DeBusk College of Osteopathic Medicine, Lincoln Memorial University, Harrogate, TN 37752, USA; Fareeha.moiz@lmunet.edu (F.M.); J.souzaperes@lmunet.edu (J.V.S.-P.); valerie.samuels@lmunet.edu (V.S.); tarun.masimukku@lmunet.edu (T.M.); Myra.khalid@lmunet.edu (M.K.); 2School of Medical Sciences, Lincoln Memoria University, Harrogate, TN 37752, USA; Michelle.heinan@lmunet.edu; 3Medical Center Long Term Care, University of Pittsburgh, Pittsburgh, PA 15260, USA; Lredmond1959@yahoo.com; 4Of the Earth Wellness, Charlotte, NC 28204, USA; health@drdave.com; 5Research Division, Herscu Laboratory, Amherst, MA 01002, USA; paulherscu@gmail.com

**Keywords:** herb drug interaction, pharmacovigilance, phytovigilance, drug interaction

## Abstract

This review examines three bodies of literature related to herb–drug interactions: case reports, clinical studies, evaluations found in six drug interaction checking resources. The aim of the study is to examine the congruity of resources and to assess the degree to which case reports signal for further study. A qualitative review of case reports seeks to determine needs and perspectives of case report authors. **Methods:** Systematic search of Medline identified clinical studies and case reports of interacting herb–drug combinations. Interacting herb–drug pairs were searched in six drug interaction resources. Case reports were analyzed qualitatively for completeness and to identify underlying themes. **Results:** Ninety-nine case-report documents detailed 107 cases. Sixty-five clinical studies evaluated 93 mechanisms of interaction relevant to herbs reported in case studies, involving 30 different herbal products; 52.7% of these investigations offered evidence supporting reported reactions. Cohen’s kappa found no agreement between any interaction checker and case report corpus. Case reports often lacked full information. Need for further information, attitudes about herbs and herb use, and strategies to reduce risk from interaction were three primary themes in the case report corpus. **Conclusions:** Reliable herb–drug information is needed, including open and respectful discussion with patients.

## 1. Introduction

Use of herbal supplements in the United States is a multi-billion-dollar industry with sales in 2019 reaching nearly ten billion dollars [1]. Several surveys have revealed that more than one-third of U.S. citizens report use of at least one herbal product [2,3,4], with many of the herb-using respondents specifying that the use of herbals demonstrates independence in self-management of their health [4,5]. Pharmacovigilance for drug–drug interactions is fraught with complexity, confounded by factors such as comorbid conditions, pharmacogenomic variations in response and metabolism, and the influence of polypharmacy [6,7]. Phytovigilance for potential herb–drug interactions is further complicated by the multiplicity of constituents in botanicals, confusion caused by use of shared common plant names, misidentification of species, mislabeling of products, contamination of botanicals, and combination of botanicals or use of multi-botanical products [8,9,10,11,12,13,14,15]. Where the botanical product is an extract, the extraction process may alter constituents, thus altering the nature or extent of interaction with pharmaceutical agents [10,14,16]. The full complement of constituents in whole plant parts may also impact pharmacodynamics and pharmacokinetics compared to extracts in what has been termed in cannabis research as “the entourage effect” [17]. For example, bioavailability of the anti-malarial compound artemisinin is reportedly 45 times greater when whole leaf of *Artemisia annua* is administered compared to administration of pure artemisinin alone [18]. Production of secondary metabolite constituents by a plant species is also variable, impacted by geography, genotype, plant part used, and seasonal variation. Harvest, preparation, and storage of crude plant may similarly impact constituent content [10,14,19].

Sources of information regarding herb–drug interactions are themselves potentially problematic. Results from in vitro studies may offer mechanistic insight but may not translate well into clinical practice [10,20,21,22]. Animal models may offer insight into the role of metabolites in interaction, but these metabolites may not always be applicable to humans and require further clinical confirmation [23]. Clinical trials are often performed in healthy homogenous adult populations, and sometimes test inappropriate plant parts or inappropriately prepared products [10,19]. Case reports in the literature often signal that further research is required and may offer pragmatic insight into the clinical nature of herb–drug interactions but may introduce reporting bias skewed toward risk rather than benefit, may overlook confounding factors, rarely establish causality, and seldom include perspectives outside the framework of the current medical culture [14]. Interaction checking databases and publications often cite results of clinical studies, in vitro findings, and case reports from the literature. These important clinical tools may reflect risks that lack clinical relevance and may fail to offer insight into combinations that offer benefit. A review published by Ng et al. [24] provides an excellent evaluation of sources of herb–drug interaction and adverse effect information.

As we continue our progress toward integrative patient-centered care, a thorough understanding of herb–drug interactions is necessary to maximize positive outcomes from beneficial interaction while minimizing risks from potentially harmful combinations. Cooperation between those who care for patients in the context of the dominant medical belief system, those who provide care through non-dominant systems, and input from the patients themselves is needed to gather the data to form valid conclusions about the risks and benefits of any individual medicinal product [14,25]. The United States FDA adverse event reporting system (FAERS) is publicly accessible, includes consumer reporting and allows for reporting of botanical precipitants in herb–drug interaction but does not separate herbs from mainstream pharmaceuticals [26]. The World Health Organization Vigibase™ [27] is a global repository of pharmacovigilance data but is not freely accessible to individual clinicians.

Ivan Stockley, an innovator in the field of pharmacovigilance for drug–drug interactions, stated that information regarding drug–drug interactions is sometimes “no more than speculative and theoretical scaremongering guesswork, hallowed by repeated quotation until they become virtually set in stone” [16] (p. 2). The overarching goal of this scoping review is to explore the current state of clinically relevant herb–drug interaction information. Specific aims include qualitative assessment of clinical case reports, evaluation of the degree to which the herbs involved in case reports appear to serve as a signal for investigation through clinical study, and comparative evaluation of the inter-source agreement of herb–drug interaction information between case reports and clinical studies, and between case reports and selected herb–drug interaction checking resources. A secondary aim is a qualitative assessment of the attitudes toward herbal medicines expressed by authors of herb–drug interaction case reports. Review of herb–drug interaction mechanisms and extensive evaluation of individual interactions are beyond the scope of this review; the reader is referred to recent reviews by Awortwe [21,28], Borse [29], Chen [22], Liu [30], and Rombola [31] for further information on these topics.

## 2. Materials and Methods

A systematic search of Medline was performed on 28 October 2020, for journal articles and editorials using the MeSH terms “Herb-Drug Interactions” and “Drug Interactions” AND “Herbal Medicines” for case reports and clinical human research studies published in English without date restriction. Exclusion criteria included review articles, ex vivo studies, animal studies, and articles without report of herb–drug interacting combinations (e.g., those reporting only adverse effects). Retrieval precision was improved through the addition of filters including “human”, “randomized controlled trials”, “letters”, “case reports”, and “editorials”. Case report documents were mined to identify additional cases.

Case reports were evaluated for completeness by at least two separate authors using the rating criteria seen in Table 1, which represents modification of a previously published reliability rating scale [32]. Discrepant ratings were resolved through discussion. A reliability index for each report was calculated by dividing the sum of scored points by the number of applicable points. Rechallenge with problematic substances is often deemed unethical and thus is often not performed. Rather than allowing an ethical decision to impact completeness scoring, where rechallenge was not performed, 0 points were added to the numerator and 1 point was removed from the denominator for that report in calculation of the report reliability index.

MaxQDA 2020 (Verbi software) [33] was used to facilitate qualitative analysis of case reports. Case reports were analyzed to identify themes regarding the authors’ perspectives and attitudes regarding herbal medications and the roles of herbal medicines in altering response to pharmaceuticals. Reactions in case reports were evaluated for severity of reaction as fatal, severe requiring hospitalization or treatment, minor requiring no treatment beyond herb or drug withholding, beneficial if the author clearly stated benefit, or none if no interaction was found. Herb–drug interacting pairs were labeled categorically based upon interaction that increased activity/caused abnormal toxicity of the target drug, lack of interaction present, or interaction decreasing activity/level of the target drug. Individual herbs from reports involving more than 5 combined botanicals were eliminated from the analysis, as this number reflects a common cut-point for polypharmacy beyond which adverse effects become more common [34,35]. For case reports involving 5 or fewer botanicals, each individual botanical was evaluated against the primary suspected target drug(s). For case reports using only the common name “ginseng”, the three most popular botanicals using this common name were each included for the inter-source analysis. Conflict in nature of interaction between multiple reports was resolved through exclusion of the score related to multi-herbal report, which resolved all conflicts.

Herbal products studied in clinical trials investigating actions of individual herbs were identified and quantified for comparison to those identified in case reports; trials exploring multiherbal formulations were excluded from the analysis. When studied herbs or constituents were involved in case reports, the pharmacokinetic pathways for drugs noted in the case reports were identified through Lexicomp [36] and/or the Pharm GKB database [37]; details are found in Appendix A. Findings of pharmacokinetic studies were matched to the relevant case report drugs via reported pharmacokinetic pathways and assigned a categorical value for findings that would be predicted to increase drug effect, for those finding no likely interaction, and for findings that would likely decrease target drug effect. Pharmacodynamic studies were similarly labeled based upon additive/synergistic findings, no interaction, or antagonistic finding. Studies demonstrating benefit were so noted; details are presented in Appendix A. Percent agreement/disagreement for case reports versus study findings was calculated by matching pharmacokinetic pathways and pharmacodynamic actions of target drug in case reports with the pathways/actions explored for the precipitating herb in clinical studies, counting matches that supported case reports, those offering no support, and those that contradicted the reported interactions, summing each group and dividing by the number of studies that provided data for each relevant pathway or action.

Herb–drug interacting pairs identified through case reports were searched for interaction in six drug interaction information sources including Integrative Pro [38], Lexicomp [36], Medscape [39], Memorial Sloan-Kettering Cancer Center [40], Natural Medicines Database [41], and Stokely’s Herbal Medicines Interactions [16]. Three of these sources were selected for their frequent use by clinical research team members and three were selected through their position when returned by Google™ search using the terms “herb drug interaction checker”; the latter to provide insight into information found in non-subscription databases. Each herb and target drug involved in the case reports involving fewer than 5 botanicals was investigated in each database and assigned a categorical value to indicate increased effect or concentration of target drug, lack of reported interaction, antagonistic interaction, lack of data regarding at least one of the interactors, or beneficial interaction. Where a database returned conflicting information (both increasing and decreasing effect for a pair), the interaction was scored specifically for the drug if the conflicting information was generic for the class or pharmacokinetic, otherwise the interaction was scored based upon strength of the supporting information presented. Warfarin was searched in databases as a proxy for the less commonly used vitamin K antagonists fluindione and phenprocoumin. Appendix A details the findings. Cohen’s kappa was calculated for each database compared to case reports as a measure of inter-source reliability using IBM SPSS statistical package [42].

## 3. Results

The initial literature search returned 1745 records. After exclusion of duplicates (n = 30) and articles published in languages other than English (n = 15), application of filters excluded an additional 1476 articles. Screening of titles and abstract led to exclusion of 104 articles, with 53 case reports and 67 clinical study documents remaining for full review. Of these, three case reports were unretrievable. Two case reports were excluded for off-target report of adverse effect rather than herb drug interaction. Mining of case reports led to identification of an additional 53 records; one was excluded as a duplicate and three for off-target reporting of adverse effect, leaving a total of 99 case reports or series covering 107 herb–drug interaction reports and 65 clinical studies involving 82 investigations of 30 different herbs or herb combinations for review. The PRISMA flow diagram is depicted in Figure 1.

### 3.1. Case Report Analysis

*Hypericum perforatum* (St. John’s wort) was the most common herb implicated as a single precipitant (n = 17), followed by *Vaccinium spp*. (cranberry n = 8), *Panax ginseng* (n = 5), *Ginkgo biloba* (n = 4), ginseng species not otherwise specified (NOS), *Lycium barbarum* (Goji), *Curcuma longa* (source of turmeric), and *Cathus edulus* (khat) (n = 3 each). Not surprisingly the most common drugs implicated are often those with narrow therapeutic indices including as a single agent warfarin (n = 30), and as classes of drugs serotoninergic (n = 14), calcineurin inhibitors (n = 13), chemotherapy and antiretroviral therapy (n = 8 each). Five reported fatalities occurred, two associated with a combination of cranberry and warfarin, one ginkgo-related bleed in a patient taking ibuprofen, a ginkgo-related seizure fatality in a patient taking phenytoin and divalproex, and a *Mitragyna speciosa*-related (kratom) case of toxic quetiapine levels. Table 2 summarizes the interactions of the most commonly involved herbs and/or interactions with fatal outcomes; further information regarding the document group titled “case reports” are found in Appendix A. It is interesting to note that kratom is implicated in 58 herb–drug interaction reports and 42 fatalities in the FAERS database, while cranberry is associated with only 9 reports, none of them fatal [26].

These discrepant findings highlight a challenge in pharmacovigilance: consistency collection of data [12,14,44,45]. Several tools are available to aid in the objective rating of the probability that an adverse drug event or drug interaction occurred, including the Naranjo Adverse Drug Reaction Probability Scale [46] and the Horn Drug Interaction Probability Scale (DIPS) [47]. In application of these tools, a point is scored for each criterion that is associated with an increased likelihood that an observed reaction is due to the product in question. A rating tool was used by the case reporter to evaluate the probability for 28 cases in the case report document set, with 5 reports as possible (DIPS n = 2, Naranjo n = 3), 24 as probable (Naranjo n = 14, DIPS n = 8, other = 2, two reporters used two scales each), while one reporter claimed use of the Naranjo scale but did not disclose the result.

In an earlier study, Fugh-Berman and Ernst [32] applied a 10-point reliability rating score to evaluate the reliability of herb drug reports in the literature. To evaluate completeness of each of the 107 case reports in this study in a standardized manner, a “reliability index” based upon the Fugh-Berman scoring was calculated for each report. In addition to the original criteria, a point was added for pharmacologic feasibility, a provision for the full scientific name of botanicals as a minimum for “adequate description of interactors” was added, and the score was not reduced when rechallenge was omitted. Figure 2 is a histogram representing the reliability index calculated for each of the 107 cases, with a maximum attainable score of 1.

#### 3.1.1. Reliability Index Results

The reliability index scores are a measure of report completeness that ranged from 0.2 to 1, with median 0.8 and mode 0.8. The point most often lacking in the reliability index score was full evaluation of alternative explanations, with 46 (43.0%) reports rated as lacking in this domain. The next most frequently lacking points included identification of comorbidities and other medications/herbals associated with adverse effects (n = 40, 37.4%), identification of concomitant medications and herbs (n = 37, 34.6%), adequate description of interactors (n = 35, 32.7%), and information regarding remission upon de-challenge (n = 31, 29.0%). In the 15 cases that involved rechallenge, 3 (20%) did not lead to re-emergence of the reaction. Inadequacy of event description (n = 4, 3.7%), inadequate description of patient demographics (n = 9, 8.4%) incompleteness of chronology (n = 8, 7.5%), lack of reason in chronology (n = 5, 4.7%), and lack of pharmacologic feasibility (n = 13, 12.1%) were deemed least problematic.

#### 3.1.2. Qualitative Analysis of Case Reports

Three themes emerged from qualitative analysis of the case report document set: attitudes toward herbs, herb use, and patient autonomy, approaches to risk mitigation, and calls for information and education.

On one end of the attitude about herbs spectrum, four documents offered positive evidence for an herbal supplement, and one offered a balanced input comparing risk from herbs and conventional drugs, thus 5 documents (5.1% of all case reports) were rated with a non-judgmental/positive attitude toward herbs. Only two reports—including the earliest record found—hinted that the interaction may be applied for benefit. On the other end of the attitude spectrum, thirteen documents (13.1%) made statements using language that was dismissive of herbs or posited that all risk was from the herb, without acknowledgment of the risk-related role of narrow therapeutic index drugs. Additionally, fourteen reports cited irrelevant herb–drug interactions to support their case. Seven reports discussed patient decision to use herbs in a disrespectful fashion.

Twenty documents (20.2%) emphasize the importance of asking the patient about herb and vitamin supplements. Several risk mitigation strategies are discussed: four authors offer strategies to reduce risk from a medication for patients who desire the herbal product(s), 10 authors suggest increased therapeutic drug monitoring, 12 authors suggest avoidance of herbal medicine, two authors call for greater regulation of herbal sales, and one author suggests monitoring of herbal consumption for patients on high-risk medications.

Information was a theme in 32 documents. Fifteen authors identified the need for education, with eight prioritizing patient education, three prioritizing provider education, and four recommending increased education for both. General lack of information was cited by six authors as problematic. Three authors suggested a need for improved labeling, two called for improved pharmacovigilance reporting, two for better electronic resources, and one emphasized the need to identify the composition of products involved in reports.

### 3.2. Case Report and Study Comparison

Thirty different herbal products were investigated in clinical studies, including six multiherbal formulations against 24 different potential interaction mechanisms. Of the tested mechanisms, 93 were relevant to interactions found in the case report document set with 91 related to pharmacokinetic interactions and two to pharmacodynamic interaction specific to an herb–drug pair reported in at least one case report.

The herbs with most frequent study excluding multiherb formulations can be seen in Table 3, which compares the number of studies and case reports to the two lists (each from a different source) of top selling herbal products in 2019 published in *HerbalGram,* the journal of the American Botanical Council [1]. Of the three herbs associated with fatal case reports, ginkgo was most frequently studied, cranberry was investigated once, kratom was not found in the clinical trial document set. A summary of clinical study materials and methods including source of botanical is found in Appendix A.

The number of herbs shared between the set of herbs in case reports (n = 61) and those studied (n = 30) was fourteen, a union representing 46.6% of studied herbs. Thirteen studied herbs are listed as best sellers; nine of these were involved in a case report.

In the analysis of agreement for the relevant studies, 52.7% of pharmacokinetic findings supported the case reports, 37.4% offered no support, and 16.5% found interaction in that opposed the nature of the case report. Pharmacodynamic studies (n = 2) offered 100% support for the relevant case-reported interactions. 

### 3.3. Case Report and Interaction Checker Comparison

Cohen’s kappa for inter-source reliability between each interaction checker and case reports are presented in Table 4. For fatal case reports, all interaction checkers agreed that ginkgo and ibuprofen have additive risk for bleed, four and three of six agree that gingko could reduce efficacy of phenytoin and divalproex, respectively, two of six agree that kratom could produce additive effects with quetiapine, and all agree that cranberry may have additive effects with warfarin.

## 4. Discussion

It is not the intent of this study to malign those who have taken the time to share their experience nor to dismiss the work that they have done to improve our knowledge of herb–drug interactions. The purpose of this study is exploration of three bodies of information to identify limitations for their clinical application and to initiate conversations that expand our understanding of this phenomenon. One must always respect the evaluation of those with first-hand knowledge of the case. Contextual information is lacking when performing a review of case reports.

Many herbs belong in a subset of the category “drugs”. Pharmaceuticals themselves often originate from natural sources. As drugs, there is no doubt that herbs may interact with other drugs or nutrients. Herbs are fortunately available to all for use. Unfortunately, this availability adds a further layer of complexity to phytovigilance: prevalence of use for any herb is difficult to discern.

During planning discussions for this project, different attitudes regarding herb–drug interactions among team members from different disciplines of medicine became apparent, which signaled a need to include a thematic investigation. The Council for International Organizations of Medical Science includes beneficial interaction in their definition of pharmacovigilance “signal” regarding drug–drug interactions [48]. In conventional medicine, interaction between pharmaceuticals is occasionally used for benefit. Examples include the use of ritonavir to “boost” other protease inhibitors via CYP 3A4 inhibition [49] and the use of probenecid to prevent cidofovir-related renal toxicity through inhibition of renal anion transporters [50]. That piperine in black pepper inhibits the p-glycoprotein transporter should not automatically be judged in a negative light; it may enhance bioavailability/effectiveness of chemotherapeutic agents and other botanical constituents via inhibition of p-glycoprotein [51,52].

Given that pharmacovigilance systems target safety more than therapeutics it is not surprising that the case reports—and findings in FAERS—are biased toward reporting harm. Two case reports hinted at the possibility of benefit from interaction, one interaction checker reported two interactions as beneficial (albeit in opposition to case report findings), and two clinical studies reported benefit from interaction. Perhaps the ability of botanical constituents to decrease pharmaceutical drug clearance could be harnessed to reduce drug costs through reduction in dose requirements, lower the burden of drug excretion into the water supply, or enhance bioavailability of nutrients [53,54]. Further exploration could conceivably improve patient quality of life through reduction in side effects [53]. While the possibility of occasional benefit from interactions exists, far too often inappropriate combinations of drugs from any source can lead to serious harm or even death.

Complementary and Alternative Medicine (CAM) can be beneficial and aid in the treatment of many diseases. When it comes to cancer treatments, roughly half of the patients receiving treatment use CAM [55]. Patients decide to use alternative medicine alongside chemotherapy for many reasons including strengthening of the immune system, improving well-being, and relieving symptoms of either the disease or disease treatment, such as nausea, insomnia, and pain [56]. Publications evidencing positive and negative effects of the use of herbs with chemotherapy abound. To expand patient options and respect patient autonomy, physicians should be encouraged to open discussion with patients, seeking input from experts in integrative medicine when needed so that we may all participate actively in informed shared decision-making [57].

Patient autonomy is one of the four principles stated by Beauchamp and Childress in 1985 to define the morals that guide medical ethics [58]. Patient choices regarding healthcare often do not align with the tenets of the current medical paradigm. Four documents contained strong language to support autonomy and respectful discussion with patients such as “we hope to inspire the discussion about the safety of any herbal formulas in combination with cytotoxic therapies and encourage physicians to seek faithful conversations dealing with the use of CAM in cancer patients” [59] (p. 3) and “it is our professional duty to increase our understanding of alternative medicine and to provide unbiased information without sounding judgmental or indifferent” [60] (p. 1653). Seven documents contained language with the opposite tone including “the fact that this patient was a registered nurse also did not prevent her from combining medications, although it is arguable that her judgement may have been subtly affected by the TBI” (traumatic brain injury) [61] (p. 363) and “the patient was informed that discontinuation of SJW (St. John’s wort) would be next if her symptoms fail to resolve” [62] (p. 683).

Eight case report documents and three clinical study documents state that herbal medicines are used due to the fallacious belief that “natural means safe”. An example found in the case report corpus states that it is a “common belief that natural plant products never do any harm to the body and that only pharmaceutical products manufactured by humans may have harmful side effects [63] (p. 219). Only one document offered a citation for this commonly stated conception about herb-user beliefs. The cited source document reported survey results where respondents reported “natural” and “safe” as separate reasons for use and concluded that this meant that users believe that “natural means safe” [64]. One must question the validity of this conclusion, especially considering the higher levels of education among herb-users in the United States [64,65,66]. This oft-cited statement may belong in the realm of Ivan Stokely’s misconceptions that become hallowed fact through the argument of repetition. Further study is needed to assess the validity of this statement.

Cohen’s kappa is a descriptive statistic that corrects for agreement due to chance when defining inter-rater reliability. Perfect congruence of rating returns a value of “1”, lack of agreement is reflected by scores close to zero, and disagreement by negative scores. [67]. None of the kappa scores reveal agreement for interaction checkers compared to case reports. This does not mean that the checkers are invalid, but rather reflects the complexity of the issue and the need for more information. The lack of agreement may equally be due to idiosyncratic reactions, contaminants, misidentification of herb. or lack of causality in case reports.

Herbs chosen for clinical studies seem to reflect the inter-related commonality of use and reports of herb–drug interaction. An agreement of more than 50% is laudable, given the potential number of confounding factors that may be present. Further study to compare reports to FAERS or use of data mining techniques as employed in pharmacovigilance [44,45,68] may yield greater insight into these relationships.

The case reports found in the literature were lacking in completeness, perhaps in part due to word count limitations associated with report venue or in part due to availability of new information gained since publication of the case [45], which impacted the score for consideration of alternate explanations. Lack of identification by scientific name is problematic, but even inclusion of the appropriate label does not guarantee that the herb was correctly identified, appropriately harvested, stored, prepared, and was contaminant-free [14].

One of the major limitations to this study is in the use of only one database to identify case reports and clinical studies, though the mining of case reports for additional cases returned a significantly higher number of cases than did additional database searches as seen in the excellent 2018 review by Awortwe et al. [21]. The risk of bias toward a particular herb is conceivably increased through reference mining. The robustness of this study would be improved through inclusion of official pharmacovigilance data, for example from FAERS, despite that none of the case report authors reported using this as a source to evaluate prior reports of interaction. The convenience sample of 3 interaction checkers may not reflect resources that are commonly used by clinicians, but likely reflect information used by those outside the clinical realm. Fifteen case report authors identified sources of information used to evaluate the reported case interaction. Most commonly Medline^®^/Pubmed^®^ [69] was used by 13 (86.6%) followed by Natural Medicines Database [41] and Embase^®^ [70] used by three (20%), IBM Micromedex^®^ [71] by two authors, and UptoDate (powered by Lexicomp^®^ [36]), Toxline^®^ [69], and Drug Interaction Facts [72] one each.

## 5. Conclusions

The identification of information as a recurring qualitative theme signals the need for improved phytovigilance that includes exploration of potential harms and benefits from combined use of herbal medications with pharmaceuticals. As seen in Appendix A, pharmaceutical drugs induce and inhibit enzymes, indicating the need to explore the role of pharmaceutics in altering response to botanic drugs. Application of data mining could expand our understanding vastly but would be critically impacted when scientific name is omitted [14,45] and would still lack needed contextual insight.

Most importantly, we need to ask, listen, and learn with respect and without bias. The author of a cranberry and warfarin interaction report said it well: “I thought I knew pretty much everything there was to know about warfarin. I had a team to help me but was not willing to ask…if you are lucky, you will learn a new nugget of knowledge or a new skill, and if you are very lucky, your attitude will improve as well.” [73] (p. 1).

## Figures and Tables

**Figure 1 medicines-08-00044-f001:**
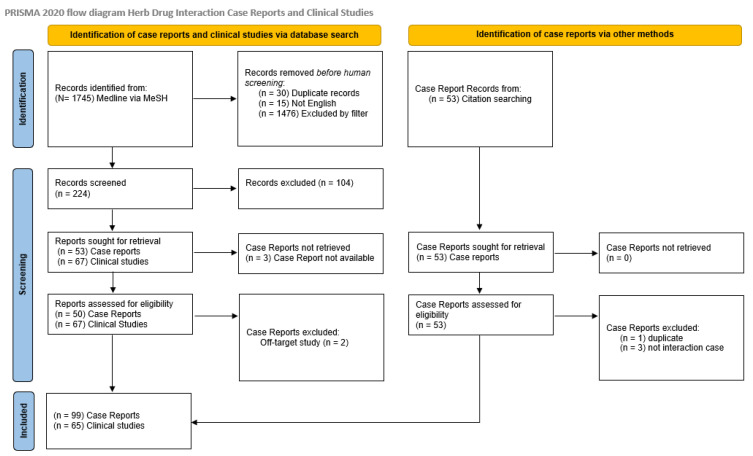
PRISMA diagram [43].

**Figure 2 medicines-08-00044-f002:**
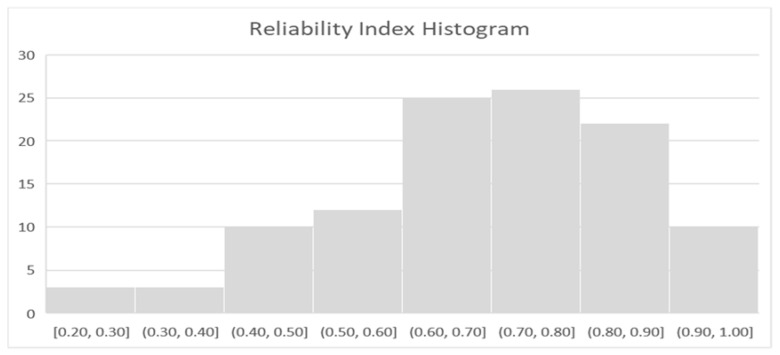
Histogram of calculated report reliability indices.

**Table 1 medicines-08-00044-t001:** Case report completeness rating tool, adapted from reference [32].

Measure	Scoring
Is reporting of relevant demographics (sex, age, relevant conditions) adequate?	Y = +1 N = 0
Are concomitant diseases and other medications associated with adverse events included (including dosing)?	
Are concomitant medications and other herbals/supplements documented (including dosing)?	
Are interactors adequately described, including scientific name of botanicals?	
Have alternate explanations been excluded?	
Is chronology complete?	
Is chronology sequence reasonable?	
Is adverse reaction adequately described?	
Is interaction pharmacologically feasible?	
Does event cease upon stopping herb?	
Does event recur upon rechallenge?	Y = +1 N = 0 N/A = 0 and remove from denominator

**Table 2 medicines-08-00044-t002:** Herbs most commonly involved and/or associated with fatal outcomes.

Latin Name, Common Name	Drug(s) or Class	N Reports	Severity (N)	Reaction
*Curcuma longa*, turmeric	Fluindione	1	Minor	Increased INR
	Tacrolimus	1	Severe	Increased tacrolimus level
contaminated with microcystin	Paclitaxel	1	Severe	Hepatotoxicity
*Equisetum arvense*, horsetail	Antiretrovirals	2	Minor	Increased viral load
*Gingko biloba,* ginkgo	Antiretrovirals	1	Severe	Treatment failure
	Ibuprofen	1	Fatal	Intracerebral bleed
	Trazodone	1	Severe	Altered mental status
	Aescinate	1	Severe	Acute kidney injury
*Ginkgo biloba* in multiherbal	Divalproate, phenytoin	1	Fatal	Seizures
	Efavirenz	1	Mild	Increased viral load
*G biloba* and *H perforatum*	Buspirone, fluoxetine	1	Mild	Serotonin syndrome
*Hypericum perforatum,* St. John’s wort	Serotoninergics	6	Mild (4) Severe (2)	Serotonin syndrome
	Calcineurin inhibitors	6	Mild (3) Severe (3)	Decreased drug levels
	Clozapine	1	Mild	Schizophrenia
	Sertraline	1	Severe	Mania
	Oral contraceptive	1	Severe	Pregnancy
*H. perforatum* in multiherbal	Cyclosporin	1	Mild	Decreased drug level
*Lycium barbarum*, goji	Warfarin	3	Mild (2) Severe	Increased INR Bleeding
*Mitragyna speciosa*, kratom	Quetiapine	1	Fatal	Increased drug level, neuroleptic malignant syndrome
*Panax ginseng*, ginseng	Phenelzine	1	Mild	Mania
	Imatinib	1	Severe	Hepatotoxicity
	Raltegravir	1	Severe	Increased drug level
contaminated with germanium	Furosemide	1	Severe	Treatment failure
*P ginseng* in multiherbal	Warfarin	1	Severe	Intracerebral bleeding
*Vaccinium* spp, cranberry	Warfarin	8	Mild (6) Fatal (2)	6 Increased INR, 2 GI and pericardial bleeding

**Table 3 medicines-08-00044-t003:** Frequency of herb study compared to frequency of case report and rank on two top selling herb lists. NOS, species not otherwise specified, multiherb, identified only as a component in a case involving multiple herbs.

Clinical Studies	N Studies	N Case Reports	Top Selling Herb Lists Ranks
*Hypericum perforatum* St. John’s wort	17	17	0, 0
*Ginkgo biloba*, ginkgo	11	7	17, 21
*Panax ginseng*, ginseng	6	4 plus 3 NOS	30, 30
*Allium sativum*, garlic	3	1 (multiherb)	8, 15
*Hydrastis canadensis*, goldenseal	3	0	0, 0
*Piper methysticum*, kava-kava	3	2 (multiherb)	0, 28
*Silybum marianum*, milk thistle	3	0	23, 10
*Actea racemosa*, black cohosh	2	0	15, 0
*Curcuma longa*, turmeric	2	3	4, 3
*Crataegus spp*, hawthorne	2	0	0, 38

**Table 4 medicines-08-00044-t004:** Cohen’s kappa and 95% confidence intervals (CI) comparing checkers to case reports.

Interaction Checker	Kappa	95% CI
Integrative pro [38]	0.011	(−0.013, 0.035)
Lexicomp [36]	0.012	(−0.012, 0.035)
Medscape [39]	0.018	(−0.007, 0.043)
MSKCC [40]	0.017	(−0.008, 0.042)
Natural Medicine Database [41]	0.014	(−0.008, 0.036)
Stockley’s Herb Drug Interaction [16]	0.009	(−0.015, 0.033)

## Data Availability

Data sharing not applicable.

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
