# Peer review of "Herb–Drug Interactions: Worlds Intersect with the Patient at the Center"

_medicines, 2021, doi:10.3390/medicines8080044_

Round 1

Reviewer 1 Report

It is a timely review of relevant reports, instructive and useful.

Author Response

Thank you for the review. We have corrected all spelling and grammar errors.

Reviewer 2 Report

The authors have prepared an extensive review of herb-drug interactions. The selected research methodology is correct, the authors use numerous databases.

Minor problems:
The paper does not present clearly the type (.or examples of the most common interactions that were analyzed. This information is included in Appendix D, but it would be valuable if this information could be provided in the body of an article or in a table in a publication. This is briefly presented in chapter 3.3, but in this type of publication it is worth to pointing it out.

Line 266: in the text "table 2", but in table "table number 3". The wrong numeration, number 2 is missing in the publication.

line 268: please check bibliography: American Botanical Council [1] does not match the bibliography list [1], but I cannot verify this.

line 306: "In conventional 306 medicine, interaction between pharmaceuticals is used for benefit." I would treat this sentence very carefully. Drug-drug and drug-herb interactions are mostly unwonted and negative and are the result of ignorance of the patients or polytherapy. The authors provide examples of simultaneous use and positive interactions, but it should be pointed, that there are also negative, dangerous interactions and they dominate, for example in the case of St. John's wort with other drugs. For being in balance, it is worth mentioning this.

Author Response

Thank you for the insightful review and suggestions. We've addressed them as follows:

The paper does not present clearly the type (.or examples of the most common interactions that were analyzed. This information is included in Appendix D, but it would be valuable if this information could be provided in the body of an article or in a table in a publication. This is briefly presented in chapter 3.3, but in this type of publication it is worth to pointing it out. - we added table 2 to offer more information on the most common herbs and those associated with fatal outcomes

Line 266: in the text "table 2", but in table "table number 3". The wrong numeration, number 2 is missing in the publication. Addition of table 2 fixed this issue

line 268: please check bibliography: American Botanical Council [1] does not match the bibliography list [1], but I cannot verify this. I reworded the sentence so that it is clear that HerbalGram is the journal of the ABC

line 306: "In conventional 306 medicine, interaction between pharmaceuticals is used for benefit." I would treat this sentence very carefully. Drug-drug and drug-herb interactions are mostly unwonted and negative and are the result of ignorance of the patients or polytherapy. The authors provide examples of simultaneous use and positive interactions, but it should be pointed, that there are also negative, dangerous interactions and they dominate, for example in the case of St. John's wort with other drugs. For being in balance, it is worth mentioning this. We clarified by adding "occasionally" to line 307 and adding a sentence  at line 331:"While the possibility of occasional benefit from interactions exists, far too often inappropriate combinations of drugs from any source can lead to serious harm or even death."

Again, sincere thanks for the review.